# Role of COMT V158M Polymorphism in the Development of Dystonia after Administration of Antipsychotic Drugs

**DOI:** 10.3390/brainsci11101293

**Published:** 2021-09-29

**Authors:** Antonio Gennaro Nicotera, Gabriella Di Rosa, Laura Turriziani, Maria Cristina Costanzo, Emanuela Stracuzzi, Girolamo Aurelio Vitello, Rosanna Galati Rando, Antonino Musumeci, Mirella Vinci, Sebastiano Antonino Musumeci, Francesco Calì

**Affiliations:** 1Unit of Child Neurology and Psychiatry, Department of Human Pathology of the Adult and Developmental Age “Gaetano Barresi”, University of Messina, 98125 Messina, Italy; antoniogennaro.nicotera@polime.it (A.G.N.); turrizianilaura@gmail.com (L.T.); 2Oasi Research Institute-IRCCS, 94018 Troina, Italy; mccostanzo@oasi.en.it (M.C.C.); emanuela.mistral@virgilio.it (E.S.); avitello@oasi.en.it (G.A.V.); rgalati@oasi.en.it (R.G.R.); amusumeci@oasi.en.it (A.M.); mvinci@oasi.en.it (M.V.); samusumeci@oasi.en.it (S.A.M.); cali@oasi.en.it (F.C.)

**Keywords:** COMT, antipsychotics drugs, dystonia, extrapyramidal symptoms

## Abstract

Antipsychotics (APDs) represent the main pharmacological strategy in the treatment of schizophrenia; however, their administration often may result in severe adverse effects, such as extrapyramidal symptoms. Typically, dystonic movements are considered the result of impaired function and/or abnormalities of dopaminergic neurotransmission/signaling in the basal ganglia. The catechol O-methyltransferase (COMT) gene is located within the 22q11.2 region, and its product is an enzyme involved in transferring a methyl group from S-adenosylmethionine to catecholamines, including dopamine. Studies showed that COMT Val158Met polymorphism modifies enzymatic activity and, consequently, synaptic dopamine concentration in specific brain areas. We identified a patient with 22q11.2 deletion syndrome presenting with cervical and trunk dystonia after paliperidone administration, which persisted even after drug discontinuation. Given the patient’s genetic condition, we hypothesized that the dopaminergic dysfunction had been aggravated by COMT involvement, thus causing dystonia. In line with this hypothesis, we carried out a study on psychiatric patients in chronic treatment with APD to evaluate the distribution of the COMT Val158Met polymorphism and its role in the onset of adverse extrapyramidal symptoms. The study included four patients with dystonia after administration of APDs compared to 17 patients who never presented adverse drug reactions. Our data suggest that the Val/Val and Met/Met polymorphisms of the COMT gene are associated with a protective effect for the development of collateral extrapyramidal symptoms in patients treated with APDs, while the Val/Met genotype could be considered a risk factor for the development of dystonia after APDs administration.

## 1. Introduction

Dopamine is a neurotransmitter that belongs to the catecholamine and phenethylamine families. It is released by neurons and plays a critical role in the motivational component, release of various hormones, and motor control. Dopamine exerts its functions by binding to different dopamine receptors located throughout the peripheral and central nervous systems [1]. Dopamine receptors are a class of G protein-coupled receptors. There are five subtypes of dopamine receptors, which include: D1, D2, D3, D4, and D5. Based on their biochemical and structural properties, these receptors are divided into two different groups: D1-like group (including D1 and D5) and D2-like group (including D2, D3, and D4). D1 receptors are post-synaptically expressed and increase the probability of neuronal firing [2], while D2 receptors are expressed both pre-synaptically and post-synaptically. When synaptically expressed, they act as autoreceptors that regulate dopamine release; when post-synaptically expressed, they reduce the likelihood of neuronal firing [3]. Moreover, D1 and D2 receptors have a different distribution in the brain and distinct transducing units. Among the D2-like receptors, there are two forms of the D2 receptors derived from alternative splicing of a common gene: D2 short and D2 long, with highest distribution in caudate/putamen and identical pharmacology. No compound has been able to distinguish between these two isoforms [1,4].

To stop signaling, extracellular dopamine is removed from the synaptic cleft and can be recycled by dopaminergic neurons or degraded by different enzymes, such as monoamine oxidase (MAO) and catechol-O methyl transferase (COMT). The COMT gene is located within the 22q11.2 region; its product is an enzyme involved in the transfer of a methyl group from S-adenosylmethionine to catecholamines, including the neurotransmitters dopamine, epinephrine, and norepinephrine. The O-methylation results in one of the major degradative pathways of the catecholamine transmitters. Its activity is crucial in brain regions with low expression of the presynaptic dopamine transporter (DAT), especially in the prefrontal cortex [5,6]. In particular, DAT levels are high in the striatum and low in the cortex, and reuptake is the predominant method of dopamine removal in the striatum, while inactivation by enzymatic degradation by COMT has a dominant role in the regulation of dopamine transmission in cortex. However, COMT inactivates dopamine more slowly than the DAT [7].

The COMT gene may present a functional polymorphism involving a valine (Val) to methionine (Met) substitution at codon 158. The Met allele is associated with reduced catabolic activity and is linked to greater tonic and reduced phasic striatal dopaminergic transmission [8]. In particular, COMT Val158Met polymorphism modifies enzymatic activity and, consequently, synaptic dopamine concentration in specific brain areas (prefrontal cortex and hippocampus) [9]. A PET study investigating D1 receptor density suggested that the Val allele was associated with lower levels activity of baseline dopamine in the cortex [10], although no associations were observed in a study on cortical D2 receptors [11].

In accordance with the classic dopamine hypothesis, antipsychotics drugs (APDs) represent the main pharmacological strategy in treating positive symptoms of schizophrenia. Moreover, different studies have shown the efficacy of several APDs, both as monotherapy and in combination with mood stabilizers, such as lithium and valproate, to treat acute mania, manic relapse prevention, and acute bipolar depression [12,13]. Despite the advent of second-generation APDs, severe adverse effects after their administration can occur, such as extrapyramidal symptoms due to excessive antagonism of D2 APDs receptor in the substantia nigra and striatum areas [14,15]. Since all APDs bind to D2 receptors, it has been suggested that blockage of these receptors in the caudate, putamen, and globus pallidus is partly responsible for causing acute dystonia [16]. Risk factors for such conditions include a history of prior extrapyramidal symptoms and high medication dose; moreover, elderly females are more susceptible to drug-induced parkinsonism and tardive dyskinesia, while, more frequently, young males manifest dystonic reactions [17].

Casarelli et al. [18] showed that COMT haploinsufficiency and dopamine activity dysfunction might influence neurological and psychiatric symptoms in 22q11 deletion syndrome (22q11.2DS), typically involved in these patients. 22q11.2DS is the most common chromosomal microdeletion disorder causing schizophrenia and movement disorders or other motor abnormalities [19,20]. Boot et al. [20] described a 22q11.2 patient affected by movement disorder and other motor abnormalities probably caused by the administration of antipsychotic drugs. It is reported that antipsychotics may cause or aggravate movement disorders in 22q11.2DS, including parkinsonism, dystonia, and dyskinesia [20,21]. Moreover, the alteration in the dopaminergic system in these patients is supported by the strong association of 22q11.2DS with early-onset Parkinson disease [22]. Finally, Kontoangelos et al. [23] described another 22q11.2DS patient who showed cervical and trunk dystonia after administering a single dose of 2 mg haloperidol. Authors hypothesized that an imbalance between dopaminergic and muscarinic receptors activity in the nigrostriatal pathway could support the role of COMT in the onset of dystonia in the patient treated.

Firstly, we identified a 31-year-old male patient with 22q11.2DS and moderate intellectual disability and schizophrenia who suffered from cervical and trunk dystonia after paliperidone administration. Once the treatment was interrupted, movement disorders persisted, thus requiring treatment with botulinum toxin. In light of the studies mentioned above, we hypothesized that the patient’s hemizygosis condition plays a fundamental role in COMT functionality, thus causing a loss of function of the enzyme and contributing to this severe side effect.

Similar to this condition, hypothesizing that the V158M polymorphism of the COMT can contribute to the onset of adverse extrapyramidal symptoms, we carried out a study to evaluate the distribution of COMT polymorphism in our psychiatric population.

## 2. Materials and Methods

The sample included Caucasian young adults/adults (age range, 15–45 years) of both genders, in psychopharmacological treatment with APDs (including polytherapy) who were admitted at Oasi Research Institute (IRCCS), Troina (EN-Italy) between December 2016 and December 2019. All patients had been receiving APDs for at least three years. Exclusion criteria included the inability to maintain a periodic pharmacological follow-up and the presence of transitory extrapyramidal adverse effects. In order to better evaluate the most resistant and the most sensitive patients, we excluded those ones who experienced only mild transient effects. Furthermore, because the distribution of the COMT v158m polymorphism is frequently found in the general population, the evaluation of the two extremes of population allowed us to precisely highlight the genotype/phenotype correlation.

The cohort of patients was divided into two groups: (i) a group with persistent dystonia after APDs administration (not responsive to treatment interruption), also called APDs-sensitive (APDS group); (ii) a group that did not present any adverse effects, identified with APDs-resistant (APDR group).

Moreover, healthy adults were enrolled as a control group.

Apart from psychopharmacological treatment, inclusion/exclusion criteria were identical to those used for the investigated patients.

The study was conducted through the review of the medical records of enrolled patients. Data were securely stored and managed using electronic data capture tools hosted at the Oasi Institute for Research on Mental Retardation and Brain Aging (IRCCS), Troina, Enna, Sicily, Italy.

### 2.1. Statistical Analysis

χ2 test was used to analyze the gene frequency Hardy-Weinberg equilibrium and compare the difference of alleles and genotypes between patients and controls, using Yates correction when necessary. The *p*-value < 0.05 was set as significant.

### 2.2. Genotyping

Genome DNA was extracted from blood by using the phenol/chloroform method. The V158M (rs4680) and L136L (rs4818) genotypes were determined by PCR and DNA sequencing. Primers were designed by software Primer3web version 4.1.0 (https://primer3.ut.ee/, accessed on 20 September 2021) and used for PCR amplification: forward (5′-GGGTGCTGCAGGAGGAG-3′) and reverse (5′-CCTGTAAGGGCTTTGATGC-3′).

## 3. Results

A total of 56 patients in chronic treatment with APDs of both genders were initially recruited, however only 21 (M:F = 13:8; age range, 16–46 years) were included in the final analysis. Fourteen patients were excluded for age criteria (<15 years of age); twelve because they did not undergo a complete psychopharmacological protocol during the follow-up, and eight for transitory adverse effects. Finally, the APDS group consisted of four patients, one of which with 22q11.2DS, while the APDR group included 17 patients. Table 1 and Table 2 show some of the patients’ features, including psychiatric diagnosis, any other associated pathology, possible dystonia-related characteristics, psychopharmacological treatment, genotypes, and allele frequencies of the polymorphisms.

The control group consisted of 56 healthy Caucasian subjects (M:F = 36:20; age range: 34–67).

### Statistical Analysis

V158M (chr22:19963748) and L136L (chr22:19963684) mapped to chromosome 22q11.2 in the same exon 4 of COMT gene, with a distance of64pb. Genotype and allele frequencies for V158M and L136L variations of APDR and control group are shown in Table 3; Chi-square test results for gene frequency Hardy-Weinberg equilibrium and alleles/genotypes discordance between APDR patients and the control group are shown in Table 4.

These results showed a significant deviation from the Hardy-Weinberg equilibrium for V158M variation in APDR patients (*p* = 0.0003), while, as we expected, the control group was in the Hardy-Weinberg equilibrium for the same variant. On the other hand, APDR patients and the control group were in the Hardy-Weinberg equilibrium for L136L polymorphism.

We hypothesized that the V158M Hardy-Weinberg deviation was mainly due to the G/A genotype in APDR patients (expected patients: 8; observed patients: 1).

This discrepancy was also evident when we compared APDR patients to controls with the Chi-square test. In fact, the analysis revealed a significant difference between the V158M genotypes (G/G, G/A, and A/A) in APDR patients versus the control group (Chi 2x2-Yates = 15.644; *p* = 0.0004; gdf = 2); this was due to the G/A genotype frequency equal to 5.88% in APDR patients, and 53.57% in the control group. Moreover, the genotype G/A revealed an exclusive variant in APDS group, while it was found just in one patient in the APDR group.

## 4. Discussion

Generally, dystonic movements are considered the result of impaired function and/or abnormalities of dopaminergic neurotransmission/signaling in the basal ganglia [24]. Typical APDs act almost exclusively as blockers of D2 receptors, while all contemporary atypical APDs are characterized by serotonin-2A antagonistic property, comparable to or even higher than their D2 blocking potential [25]. In any case, it is well known that the use of APDs can frequently induce extrapyramidal symptoms [26]. As reported by Kamishima et al. [27], approximately 7.2% of patients treated with long-acting parenteral risperidone developed acute dystonic reactions. On the other hand, according to a study carried out by María Concepción Moreno-Calvete [28] on a group of 28 patients (aged > 18 and treated with typical and atypical neuroleptics for more than six months), it was found that 21.4% of them showed extrapyramidal effects (akathisia, parkinsonism, and tardive dyskinesias). In these patients, young age, male gender, history of substance abuse, and family history of dystonia were considered as risk factors for acute dystonia [29]. 

Although pharmacogenetic studies identified the possibility of genetic risk factors (e.g., PPP1R1B, BDNF, DRD3, DRD2, HTR2A, HTR2C, COMT, MnSOD, CYP1A2, and RGS2) thus ascertaining the individual differences in response to antipsychotics, Bakker et al. [30] did not reveal a single nucleotide polymorphism associated with susceptibility to drug-induced movement disorders.

The haploinsufficiency of COMT in 22q11DS is associated with a low enzymatic activity and, consequently, high dopamine levels in the central nervous system. This can cause or aggravate extrapyramidal symptoms in these patients (including parkinsonism, akathisia, dystonia, and dyskinesia) [18]. Therefore, studies suggest that a common single nucleotide polymorphism at codon 158 in the COMT gene has been associated with differential COMT function and differential cortical synaptic dopamine accumulation in specific brain area regions (e.g., cortex and striatum). Specifically, homozygosity for Met (A/A) allele of the val158met SNP (rs4680) is associated with relatively lower COMT enzymatic activity and a more optimal D1/D2 balance, while homozygosity for the Val (G/G) allele is associated with higher activity and with a low D1/high D2 state. Moreover, individuals with Met/Val genotype (A/G) have a middle level of COMT activity, with D1/D2 intermediate [31,32]. Moreover, it is well known that estrogen is a regulator of COMT promoter activity, and COMT activity in blood is lower in females; this could influence COMT activity [33].

V158M polymorphism is frequently reported in the healthy population not requiring antipsychotic drugs (European dbSNP Allele Frequency = 0.508); however, it is assumed that the alteration of this fragile balance can contribute to the development of extrapyramidal symptoms.

The analysis of the L136L polymorphism of the COMT gene in APDR patients showed no significant difference both in the Hardy-Weinberg balance (*p* = 0.497) and in the Chi-square test analysis (*p* = 0.653) (Table 4) compared to the control group. Therefore, a balance of allelic and genotypic frequencies was found which do not differ from the control population. On the other hand, the role of the V158M polymorphism (different variant of the same COMT gene) is strengthened, thus suggesting that the overall Hardy-Weinberg imbalance (*p* = 0.0003) and the statistically significant comparison with the control population (*p* = 0.0004) can derive from a real biological basis.

In our study, all patients included in the APDS group (*n* = 4) experienced dystonia as they were carrying V158M polymorphism. In total, 94.11% of patients included in the APDR group (*n* = 16/17) did not experience any adverse effect since they were carrying A/A and G/G polymorphisms. 

## 5. Conclusions

These data suggest that G/G and A/A genotype polymorphisms of COMT gene are associated with a protective effect for developing collateral extrapyramidal symptoms in patients treated with APDs; on the other hand, the G/A genotype, almost exclusively present in APDS patients, could be considered as a risk factor for developing dystonia after administration of APDs.

In these patients, the possible manifestation of extrapyramidal effects is due to an alteration of the balance of the dopaminergic activity in specific brain areas, in particular where COMT is more expressed (i.e., prefrontal cortex). Following the same theory, the dopaminergic balance in patients with A/A and G/G genotype could confer a protective effect. Certainly, the dopaminergic system is a very complex pathway, and it can be assumed that other genetic and epigenetic factors could contribute to the development of extrapyramidal effects in this cluster of patients. Nevertheless, in line with our data and the scientific literature mentioned above, we hypothesized that COMT polymorphisms could play a crucial role in potential extrapyramidal adverse reactions associated with APDs.

In recognizing the small sample size of the study, we consider this as a preliminary hypothesis. Should these data be confirmed in further studies, this could have practical application in choosing the most suitable treatment for some groups of patients, in particular to prevent potential extrapyramidal symptoms. Moreover, in patients with bipolar or manic disorders, the presence of COMT V158M polymorphism could lead to choosing different therapeutic options rather than APDs, like antiepileptics or lithium salts.

## Figures and Tables

**Table 1 brainsci-11-01293-t001:** Characteristics of APDs-resistant (APDR) group.

Case	Age (y.o.)	Gender	Psychiatric Diagnosis	Dystonia	Associated Medical Conditions	Genetic Analysis	COMT *Val158Met* Polymorphism	Psychopharmacological Treatment
1	22	Male	Borderline Intellectual Functioning (BIF), Psychotic Disorder	Absent	Myopathy	Array-CGH: normal	Val/Val	Risperidone
2	25	Male	Severe Intellectual Disability, Psychotic Disorder	Absent	Not reported	Array-CGH: normal Genetic thrombophilia: MTHFR Gene Variants C677T Heterozygous	Val/Val	Quetiapine Risperidone
3	17	Male	Mild Intellectual Disability, Psychotic Disorder	Absent	Scoliosis	Array-CGH: normal FMR1 analysis: normal	Val/Val	Risperidone Valproate Levomepromazine
4	27	Female	Moderate Intellectual Disability, Psychotic Disorder	Absent	Not reported	Array-CGH: normal	Met/Met	Perphenazine Amitriptyline Valproate Aripiprazole Levomepromazine N-acetylcysteine
5	22	Male	Mild Intellectual Disability, Mood Dysregulation Disorder	Absent	Not reported	Array-CGH: arr16p12.2(21,599,687–21,739,885) × 1	Val/Val	Methylphenidate Risperidone Levothyroxine Sodium
6	18	Male	Severe Intellectual Disability, Psychotic Disorder	Absent	Not reported	Array-CGH: normal FMR1 analysis: Intermediate (45–55 CGG)	Met/Met	Olanzapine Risperidone
7	23	Female	Mild Intellectual Disabilities, Schizophrenia Spectrum	Absent	Not reported	Array-CGH: arr[hg19] 7p12.1(51,700,630–52,568,971) × 3	Met/Met	Valproate Olanzapine N-acetylcysteine
8	22	Female	Severe Intellectual Disability, Psychotic Disorder, Gait disorder	Absent	Primary muscle disease sign	Array-CGH: normal DiGeorge Syndrome/VCFS: normal	Val/Met	Olanzapine Topiramate N-acetylcysteine
9	18	Female	Moderate Intellectual Disability, Schizophrenia	Absent	Not reported	Array-CGH: normal	Val/Val	Risperidone Clozapine
10	20	Female	Mild Intellectual Disability, Psychotic Disorder	Absent	Not reported	Array-CGH: normal	Met/Met	Risperidone Oxcarbazepine
11	22	Male	Mild Intellectual Disability, Psychotic Disorder	Absent	Not reported	Array-CGH: normal	Val/Val	Valproate Risperidone
12	17	Female	Mild Intellectual Disability, Psychotic Disorder	Absent	Aspecific white matter alteration in brain MRI	Array-CGH: normal	Met/Met	Risperidone
13	19	Male	Schizophrenia Spectrum, Specific Learning Disorder	Absent	Cerebral atrophy in brain MRI	Array-CGH: Arr[hg19] 6q12(66,158,720–66,369,429) × 3 mat	Met/Met	Clozapine Olanzapine Risperidone
14	18	Female	Moderate Intellectual Disability, Schizophrenia Spectrum	Absent	Not reported	Array-CGH: normal. NAA15 gene: truncating variants c.239_240delAT (p.H80RfsX17)	Val/Val	Aripiprazole Risperidone Olanzapine
15	29	Male	Severe Intellectual Disability, Schizophrenia	Absent	Not reported	Array-CGH: normal	Val/Val	Risperidone Fluvoxamine
16	17	Male	Unspecified Intellectual Disability, Unspecified Neurodevelopmental Disorder	Absent	Not reported	Array-CGH: normal	Val/Val	Risperidone
17	23	Male	Schizophrenia	Absent	Not reported	Array-CGH: normalv	Val/Val	Risperidone Levomepromazine

**Table 2 brainsci-11-01293-t002:** Characteristics of APD-sensitive (APDS) group.

Case	Age (y.o.)	Gender	Psychiatric Diagnosis	Dystonia	Associated Medical Conditions	Genetic Analysis	COMT *Val158Met* polymorphism	Psychopharmacological Treatment
18	31	Male	Moderate Intellectual Disability. Schizophrenia	Focal dystonia (Cervical-trunk)	Diabetes mellitus DiGeorge’s syndrome	Array-CGH: arr 22q11.21(18,919,942–21,440,514) × 1 dn	-	Paliperidone * Aripiprazole * Venlafaxine Delorazepam
19	29	Male	Schizophrenia	Focal dystonia (foot)	Not reported	Array-CGH: normal Genetic thrombophilia: Factor II homozygous WT/WT G20210A (G/G); Factor V Leiden G1691A (G/G)	Val/Met	Risperidone * Levomepromazine *
20	20	Male	Moderate Intellectual Disability, Schizophrenia	Hemidystonia	Not reported	Array-CGH [hg19] 3p26.2(2,854,929–3,147,222) × 3	Val/Met	Risperidone * Aripiprazole Tetrabenazine Quetiapine Clozapine Clotiapine ** Lamotrigine Olanzapine Delorazepam Clonazepam Paroxetine Topiramate Fluvoxamine Levomepromazine
21	46	Female	Mild Intellectual Disabilities, Schizotypal Personality Disorder	Focal dystonia (foot)	Irsutism	Panel SCA1, SCA2, SCA3, SCA6, SCA7, SCA8: normal	Val/Met	Olanzapine Thioridazine Diazepam Carbamazepine

* Dystonia onset. ** dystonia exacerbation.

**Table 3 brainsci-11-01293-t003:** Genotype and allele frequencies of V158M (rs4680) and L136L (rs4818) polymorphisms in COMT gene of APDR patient (*n* = 17) and control group (*n* = 56).

SNP	Genotypes	Frequency (%)		Allele	Frequency (%)	
		APDR Patients	Controls		APDR Patients	Controls
V158M	G/G	58.82	28.57	G	76.47	55.36
	G/A	5.88	53.57	A	23.53	44.64
	A/A	35.29	17.86			
L136L	G/G	23.53	17.86	G	41.18	42.86
	G/C	41.18	50.00	C	58.82	57.14
	C/C	35.29	32.14			

**Table 4 brainsci-11-01293-t004:** Chi-square test results for Hardy-Weinberg equilibrium of gene frequency and alleles/genotypes discordance between APDR patients and control group.

V158M			
Hardy-Weinberg	Chi=	*p*=	gdf=
APDR—Patients	13.029	0.0003	1
Controls	0.394	0.530	1
APDR Patients Vs Controls-Alleles	Chi=	*p*=	gdf=
Yates’s chi-squared test	0.428	0.512	1
Chi-Square test	0.873	0.350	1
APDR Patients Vs Controls-Genotype	Chi=	*p*=	gdf=
Yates’s chi-squared test	15.644	0.0004	2
Chi-Square test	19.660	0.00005	2
L136L			
Hardy-Weinberg	Chi=	*p*=	gdf=
APDR Patients	0.462	0.497	1
Controls	0.024	0.876	1
APDR Patients Vs Controls-Alleles	Chi=	*p*=	gdf=
Yates’s chi-squared test	0.009	0.034	1
Chi-Square test	0.923	0.854	1
APDR Patients Vs Controls-Genotype	Chi=	*p*=	gdf=
Yates’s chi-squared test	0.156	0.653	2
Chi-Square test	0.924	0.721	2

## Data Availability

Data available on request due to restrictions e.g., privacy or ethical.

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
