# Peer review of "Role of COMT V158M Polymorphism in the Development of Dystonia after Administration of Antipsychotic Drugs"

_brainsci, 2021, doi:10.3390/brainsci11101293_

Round 1
Reviewer 1 Report
This study evaluated the distribution of COMT rs4680 and rs4818polymorphisms in psychiatric patients with different diagnoses and assessed its role in the onset and persistence of dystonia in the same population. Many studies have already addressed this polymorphism, particularly rs4680, in different psychiatric populations including schizophrenia, and meta-analyses exist. Therefore, the data presented in table 3, regarding the distribution of rs4680 and rs4818 polymorphisms between 17 patients with different diagnoses and 56 controls, is not informative.
The other aim has focused on persistent dystonia after the cessation of antipsychotics. Out of 56 patients included, only 21 patients completed the analysis. Four patients had persistent dystonia. Three out of those 4 patients had Val/Met genotypes, while the fourth patient had 2q11.2DS, whereas two patients also had comorbid conditions.
In conclusion, the authors hypothesized that COMT polymorphisms could play a crucial role in potential extrapyramidal adverse reactions associated with ADPs. This study has focused only on dystonia. Moreover, Val/Met genotype is detected in about 40-50% participants from European population, including psychiatric patients, and majority of them do not develop persistent dystonia. Therefore, such conclusion is not supported by the data. The study sample is to small to draw any conclusion regarding the differences in genotype distribution. This article, in my opinion, may be published only as a case series of 4 patients with persistent dystonia.
Minor issues
Introduction
Lines 47-51 – please, distinguish D2 receptors, and D2-like receptors, since they are not identical. Provide short descriptions on D2-like, and than od D2 receptors
Line 68-associated with lower levels of baseline dopamine... please add what, activity, concentration...?
Line 77- Boot et al... please, add year
Results-in table 1, there are other genetic data, such as MTHFR Gene Variants, which are not mentioned anywhere in the text
Reviewer 2 Report
In this article, the authors attempted to explore the role of a COMT polymorphism in the occurence of extrapyramidal/dystonic adverse effects upon administration of antipsychotics. Although this research has merit in terms of precision medicine and finding populations at risk of such side-effects, the overall work presented is not of the highest quality, and the article is also rather sloppily written, with abundant linguistic errors. As such, I cannot recommend publication in its current form, but would allow for a second chance upon improvements.
In greater detail:
-ABSTRACT: The abstract fails to deliver the results of the study accurately, as no numbers and data are presented, not even concisely. Also, please add the role of COMT in the cleansing of excess neurotransmitters (and thus why the mutation can lead to dystonia).Finally, the DS abbreviation needs to be explained.
-Note: Shouldn't antipsychotic drugs translated to APDs, and not ADPs, as an abbreviation?
INTRODUCTION: It has some points where no clear connection between sentences is evident. For instance, the authors refer to the drugs as the main options for schizophrenia, then phrase a theory of the occurence of the dystonia, and THEN, they refer some more uses of the drugs. There is a mental gap that hinders the free flowing of the paragraph. Also, add the word "receptors" after "Dopamine receptors are a class of G protein-coupled", and change "The O-methylation results in" to "consists one of...". Finally, the case report needs significant rephrasing.
-METHODS: Why were patients with transitory extrapyramidal adverse effects" excluded if the point of the study was to assess the onset of extrapyramidal adverse effects? Only persistent dystonia was included, and why? The authors need to be more specific on the aims of the study. Also, what do the authors mean with "suspect adverse effects"? Importantly, why was the L136L (rs4818) added to the protocol, when it has never been mentioned again? Please clarify and epxlain. Also change the "ii) a group who not presented a" to "a group that did not present a..". Additionally, the denotation of the groups as hypersensible and super resistant is...weird. Maybe the authors could name the groups as "APD-S" in sensitive, and "APD-R" as in resistant, and adjust the manuscript as such.
-RESULTS: Besides having the tables, you need to have the numbers in the text, and the percentages in each group. Also, in table 2, why is the genotype of the first patient not available?If the genotypng procedure failed, you need to report the genotyping rate of your cohort. Finally, a patient cannot be "reporting" such a syndrome, please rephrase line 133, maybe say "was diagnosed with" or something similar.
-DISCUSSION: I feel that the discussion is rather short and weak. Authors should better tie the presumed enzymic function of the polymorphisms to the occurence of dystonia, so that readers may easily understand why carriers of a certain genotype develop the extrapyramidal effects. Additionally, the paper fails to explain why BOTH val/val and met/met act as protective genotypes, when they authors just said, before this statement, that these genotypes lead to opposite enzymic functions and receptor states. Additionally, facts and results mentioned in the limitations of the study should be brought to the general discussion, of associating this study's results to those of previous studies on the subject. As such, the limitation section could be removed as a subheading, so that the discussion stays homogeneous. The reporting of the results of Han et al. is also very badly written.
-CONCLUSIONS: Given the number of the involved patients and the overall strength of the study, authors should highlight that the insights they provide are preliminary at best.
-OTHER POINTS: Linguistic revision is more than recommended, as some grammatic and syntactical errors can be found, besides the typos. Some of the errors have been pointed above for the faciliation of the authors. Other include for example: line 60, dominant role, not dominates, proper use of definite/indefinite articles, line 77 add the DS after the polymorphism, the exclusion criteria and the "the failure retirement of defined age" need extensive rephrasing, etc. Also, the table labels need changing, they should not be written as "Table shows X", but rather have a title as "Characteristics of XX patients"
Round 2
Reviewer 1 Report
The authors have accepted most of the comments.
Author Response
We would like to thank the reviewer for taking the time and energy to help us improving the manuscript
Reviewer 2 Report
The authors have adequately replied to most of the comments made. Only minor corrections are left, and I have no objection with the manuscript being published:
-In the new introduction, the first paragraph, as I had also mentioned before, the word "receptors" needs to be added after the "class of G protein-coupled", to read "G protein-coupled receptors". Also, in the 3rd paragraph, its beginning should be "THE COMT gene", not just "COMT gene"
-Methods: "the inability to periodic pharmacological follow-up" should be "the inability to maintain a periodic...". I would also like to see the reasoning for excluding the transitory effects, that you mentioned in the letter, in the article as well.
-Discussion: "patients treated with long-acting parenteral risperidone developed acute dystonic reactions)": please delete the bracket at the end of this sentence.
